# Retrieval Augmented Zero-Shot Text Classification

## ABSTRACT

Zero-shot text learning enables text classifiers to handle unseen classes efficiently, alleviating the need for task-specific training data. A simple approach often relies on comparing embeddings of query text to those of potential classes. However, the embeddings of a simple query sometimes lack rich contextual information, which hinders the classification performance. Traditionally, this has been addressed by improving the embedding model with expensive training. We introduce QZero, a novel training-free augmentation approach that reformulates queries by retrieving supporting categories from Wikipedia to improve zero-shot text classification performance. Our experiments across six diverse datasets demonstrate that QZero enhances performance for state-of-the-art static and contextual embedding models without the need for retraining. Notably, in News and medical topic classification tasks, QZero improves the performance of even the largest OpenAI embedding model by at least 5% and 3%, respectively. Acting as a knowledge amplifier, QZero enables small word embedding models to achieve performance levels comparable to those of larger contextual models, offering the potential for significant computational savings. Additionally, QZero offers meaningful insights that illuminate query context and verify topic relevance, aiding in understanding model predictions. Overall, QZero improves embedding-based zero-shot classifiers while maintaining their simplicity. This makes it particularly valuable for resource-constrained environments and domains with constantly evolving information.[1]

## CCS CONCEPTS

• **Information systems → Generate the Correct Terms for Your Paper**.

## KEYWORDS

Retrieval augmented learning, Query reformulation, Zero-shot text classification, Text embeddings

**ACM Reference Format:**
Anonymous Author(s). 2018. Retrieval Augmented Zero-Shot Text Classification. In . ACM, New York, NY, USA, 9 pages. https://doi.org/XXXXXXX.XXXXXXX

---

[1]code available at :https://anonymous.4open.science/r/QZERO-566F

---

*Conference'17, July 2017, Washington, DC, USA*
© 2018 Copyright held by the owner/author(s). Publication rights licensed to ACM.
ACM ISBN 978-x-xxxx-xxxx-x/YY/MM
https://doi.org/XXXXXXX.XXXXXXX

## 1 INTRODUCTION

Zero-shot learning enables classifiers to handle unseen classes efficiently, alleviating the need for task-specific training data. Unfortunately, supervised classification models encounter significant challenges in scenarios that are characterized by an unconstrained label space [1, 7]. For example, consider the task of classifying recipes into categories based on their ingredients or regional cuisines. The diversity of culinary traditions and the continuous emergence of new dishes create an expansive and evolving label space. This complexity makes it difficult to define and collect labeled data for all possible label types, thus limiting the model's ability to accurately classify novel or unique recipes. The conventional practice of retraining a model for each new label set becomes impractical due to the substantial increase in expenses associated with annotation and computation. This has necessitated the widespread adoption of zero-shot text classification.

Generative Large Language Models (LLMs) have revolutionized the field of natural language processing [3, 23], especially for zero-shot learning. Their remarkable zero-shot abilities enable them to tackle diverse tasks with impressive efficiency and adaptability. However, applying generative LLMs directly for zero-shot text classification tasks presents challenges due to their massive size and computational demands. These models require significant computational resources for inference, which can render them impractical in resource-constrained settings. In addition, these models tend to make predictions that are independent of user-specified classes [35]. For instance, when classifying cuisine as Chinese or Mexican, a generative LLM model might incorrectly predict Italian, even though it wasn't an option. This lack of user control over the classification process poses a significant limitation to achieving targeted results.

A straightforward and efficient alternative for zero-shot text classification involves assigning a class (or label) to a query (or text) by comparing the embeddings of the query and potential classes using a distance metric such as cosine similarity [15, 27]. This relatively cheap approach removes the need for retraining or additional data labeling and allows for control over the classification process. However, when applied to queries that do not explicitly reflect the context of the class or align well with the model's training data, the accuracy of this technique decreases. For example, consider Table 1, which illustrates the contrast between explicit and implicit queries for a text input whose ground truth class is Technology. In the explicit example, keywords like "Artificial Intelligence" directly suggest Technology as the correct class. In contrast, the implicit example does clearly indicate the ground truth class. Here, a model's ability to infer Technology heavily relies on prior knowledge stored within its representations. For example, if the model lacks prior knowledge that "Sam Altman" and "Anthropic" are related to technology, it will fail to generate appropriate embeddings. Consequently, queries lacking sufficient contextual cues will result in reduced accuracy and recall. [20].

**Table 1: Explicit and Implicit query formats example for a query with the ground truth Technology. In the Explicit query, the keyword Artificial Intelligence may directly suggest Technology as the potential label.**

| Query Type | Definition | Example |
|---|---|---|
| Implicit | The intended class category is implied but not stated | Sam Altman to discuss possible collaboration with Anthropic |
| Explicit | The intended class category is clearly stated | Sam Altman, the CEO of a popular Artificial Intelligence Tech company, is set to discuss possible collaboration with Anthropic, a significant competitor |

We introduce QZero, a simple retrieval augmentation approach to enhance the quality of embeddings used in zero-shot classification. Retrieval systems have emerged as powerful, cost-effective tools for various knowledge augmentation tasks, offering access to relevant information to keep models updated [12, 33, 24, 1]. They also benefit tasks like evidence-based modeling and text generation by expanding vocabulary and facilitating domain adaptation. Despite their potential, retrieval models remain under-explored in improving the quality of embeddings in zero-shot text classification tasks. The QZero paradigm explores reformulating queries by retrieving supporting information from Wikipedia. This approach enhances the quality of input text and achieves better zero-shot classification performance without model retraining. QZero streamlines the classification process by ensuring the embedding is enriched with a diverse vocabulary while remaining current with minimal resources.

Specifically, QZero, adopts a two-step approach for zero-shot classification tasks, as illustrated in Figure 1. First, it utilizes a retrieval model to identify relevant categories within supporting documents for the input text. These retrieved categories are then reformulated as the new input, depending on the type of embedding model used. For contextual embedding models, the concatenation of the categories serves as the reformulated input. Static word embedding models, on the other hand, utilize keywords and frequencies extracted from the retrieved categories. Consequently, the model utilizes this reformulated input to generate embeddings for downstream text classification tasks.

We evaluate QZero on six diverse datasets, using embedding models of different sizes, ranging from a simple Word2Vec to Open AI's text embeddings. Our results demonstrate that QZero benefits models of all scales. Notably, the additional context provided by QZero acts as a knowledge boost for smaller models, allowing them to achieve performance levels comparable to larger models. This translates to significant computational cost savings, as smaller models require less processing power. In addition, QZero offers meaningful insights that illuminate query context and verify topic (class) relevance, aiding in understanding model predictions.

## 2 RELATED WORK

### 2.1 Zero-shot Text Classification with Generative LLMs

Zero-shot classification has seen significant advancements through various approaches. One prominent avenue utilizes large-scale generative models, like GPT (Generative Pre-trained Transformer) [3], for inference tasks. Trained on massive text datasets, these models excel at understanding and generating natural language, making them attractive for zero-shot classification tasks. However, their immense size and computational demands limit their accessibility and efficiency for resource-constrained environments. In addition, these models tend to make predictions outside the scope of user-defined classes, hindering the model's applicability in scenarios where class-based predictions are required.

### 2.2 Zero-shot Text Classification via Semantic Comparison

Beyond large generative models, an alternative approach leverages embedding techniques to compare semantic similarity between the text and potential classes during classification. This approach involves encoding textual information, like words or sentences, into vector spaces using innovative methods like [21, 2, 29, 25]. Pioneering work by [5] and [8] laid the groundwork for this strategy. However, early embedding techniques had limitations in accuracy because of their reliance on simple modeling approaches, hindering their ability to effectively capture the nuanced relationships and semantic context within textual data. Recent research by [7] and [15] addressed this by fine-tuning pre-trained models with external knowledge sources like Wikipedia to improve the embedding quality for zero-shot classification. Although cheaper, these methods don't entirely eliminate the need for retraining, which can be challenging for rapidly evolving domains.

### 2.3 Retrieval Augmented Learning

To alleviate the need for retraining, recent studies like [28, 31] have explored retrieval-augmented approaches. These methods leverage external information by retrieving documents relevant to the query at inference time. These retrieved documents are then prepended to the query itself, serving as a form of query expansion. This approach improves the performance of language models on various tasks without further training the entire model. In addition, retrieval systems excel at handling new information efficiently, allowing for quick updates when new information arrives. Inspired by these studies, we investigate whether a similar retrieval strategy can improve the performance of text embedding models for zero-shot text classification tasks. Notably, while retrieval has been explored for text embeddings before (e.g., ERATE [24]), our objectives differ. ERATE focuses on reducing the high computational cost incurred from generating embeddings using large models. In contrast, QZero prioritizes enriching query context for improved classification.

### 2.4 Query Enrichment and Expansion

Query enrichment, the process of reformulating or augmenting a query with supplementary information, has been widely explored in Natural Language Processing (NLP) tasks as a means to enhance

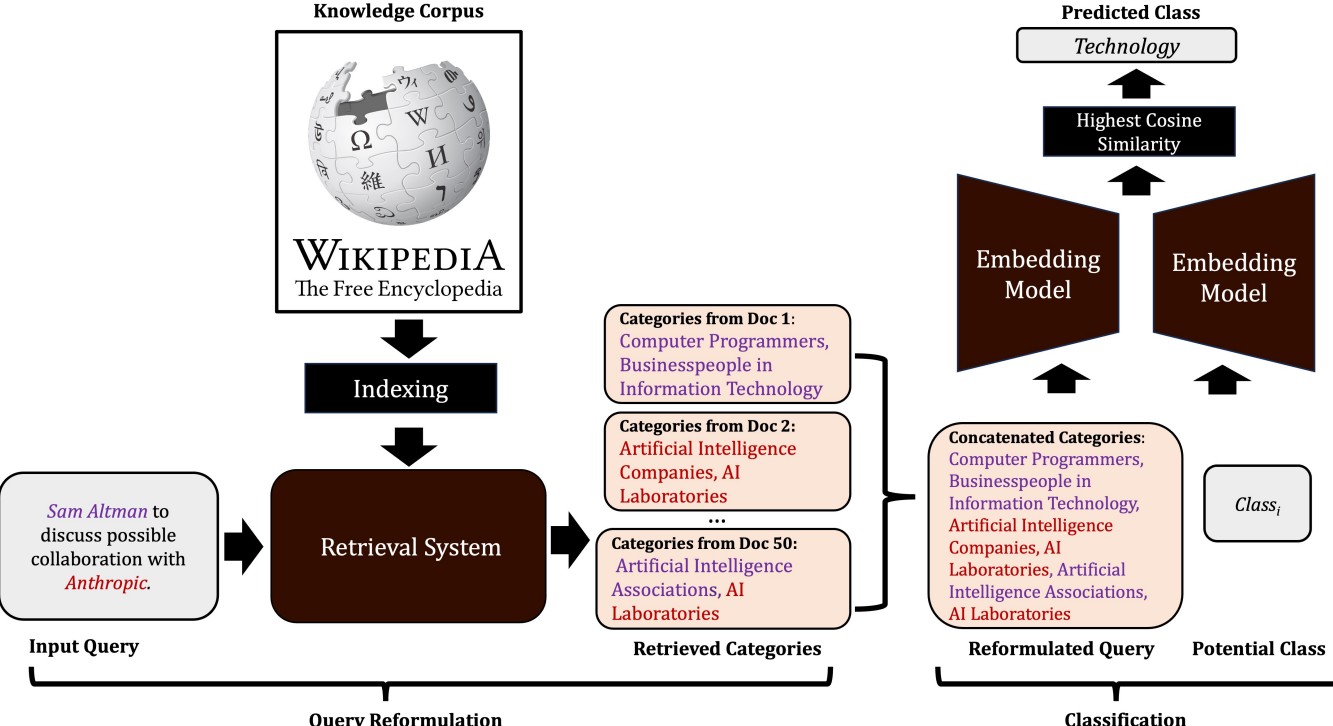

**Figure 1: Overview of the Query Reformulation Pipeline. QZero. Before classification, given a raw input query, QZero generates a new refined query by retrieving categories of Wikipedia articles that best match the original input query. The parts of the input query highlighted in purple and red represent entities that may require context information. The refined query provides the context information that may be useful for the classification process.**

semantic understanding and overall performance. While earlier research predominantly concentrated on its utility in information retrieval tasks [37, 6, 32, 14, 4], recent investigations have extended its applicability to few-shot [9] and short-text classification tasks [34]. Techniques leveraging linguistic resources like synonyms, knowledge graphs, and pseudo-relevance feedback have shown promising outcomes. Our work builds upon these prior studies by investigating the application of query enrichment specifically for zero-shot classification tasks.

## 3 METHODOLOGY

### 3.1 QZero: The Retrieval Augmented Query Reformulation Pipeline

We present QZero, a simple retrieval augmented query reformulation pipeline designed to enhance the zero-shot classification performance of embedding-based models. This system enhances the representation of essential information by transforming input text into refined queries, resulting in richer embeddings. Figure 1 provides an illustration of the QZero pipeline. Consider the input example in the figure: "Sam Altman to discuss possible collaboration with Anthropic." The input text becomes a query to the retrieval system, which returns a list of relevant document categories. The first set of categories pertains to information about Sam Altman, while the second set describes Anthropic. Afterward, the original

input is disregarded, and the reformulated query becomes the concatenation of the retrieved categories. Subsequently, the embedding of the reformulated query is compared to the embeddings of the potential classes using cosine similarity. The class with the highest cosine similarity becomes the model's prediction. In this section, we describe the components of the QZero pipeline in detail.

### 3.2 Knowledge Corpus

We used Wikipedia[2] as our knowledge corpus across all experiments. Wikipedia articles offer concise information on a wide range of subjects, organized into categories located at the bottom of the page. These categories serve as topics and keywords associated with an article. We indexed English Wikipedia articles, focusing on article content and categories. Articles with fewer than 20 words and those with no assigned categories were excluded, resulting in 5.85 million documents with at least one category.

### 3.3 The Retrieval System

The QZero scheme can be used with arbitrary retrieval systems. To build our retrieval system, we constructed a one-time index of the selected Wikipedia articles, leveraging it for subsequent retrieval of relevant article categories. Each article, represented as a unique document, contains attributes such as content and

---

[2]We downloaded wiki version: enwiki-20230820-pages-articles-multistream.xml.bz2

categories. For classification tasks, the retrieval process focuses solely on returning the categories of the best-matched articles, ensuring a concise presentation of information.

## 3.4 Keyword Extraction

Wikipedia category names are usually lengthy and more detailed than conventional label names, which poses a challenge for static word embedding models. For such models, we used different keyword extraction strategies to harness the extensive information encapsulated in the Wikipedia category as described below:

- **SpaCy's POS tagging**[3]: it identifies keywords within a sentence by breaking down the sentence into tokens. then it employs a trained statistical model to analyze each token and assign it a Part-of-Speech (POS) tag. We selected only Noun tokens as keywords for our experiments.

- **Capitalization**: SpaCy's POS tagging struggles with proper nouns representing nationalities (e.g., Indians, Filipinos). To improve accuracy with such datasets, we switched to a simpler method based on word capitalization using simple Regex functions.

- **MedCAT [16]**: When dealing with a specialized domain like medicine, we used the Medical Concept Annotation Tool. (MedCAT) for more precise extraction of medical terms and sentences. MedCAT is an open-source tool that was pretrained to identify medical terms in sentences and link them to standardized medical vocabularies.

## 3.5 Query Reformulation

QZero takes a query $x$ as an input to the retrieval system (see Figure 1). The system ranks Wikipedia articles based on their relevance to the query and returns the categories associated with the top-ranked articles. We select the categories of the top 50 articles, ensuring comprehensive coverage of potentially relevant information. Each article has a varying number of categories, and similar categories might appear across multiple articles. Our approach retains all categories from each retrieved article and we also keep repeated categories to emphasize their potential importance.

We explore the use case of the reformulated queries for contextual models that are optimized for generating meaningful embeddings from sentences and static word embedding models optimized for word inputs (details in Section 3.6). The reformulated query for the contextual embedding models comprises a concatenation of all the retrieved categories. This concatenation process is executed following the order of their respective articles' ranks, as depicted in Figure 1 and outlined in the equation below:

$$x_R^s = (C_1, C_2, \ldots, C_n)$$

In the above equation, $C_1$ represents the retrieved categories for an article. Post-retrieval, the categories for each article are denoted as $C_i$, where i signifies the rank of the article (1 for the top-ranked article). Therefore, the reformulated query $x_R^s$ is the concatenation of the retrieved categories in the sequence of their corresponding article's rank.

While contextual embedding models offer advantages, they have limitations regarding the number of input tokens. Concatenating too many categories can introduce noise into the reformulated query, and processing longer queries increases computational cost. To address these limitations, we restrict all contextual embedding models to use only the first 512 tokens of the concatenated categories as their refined queries.

Static word embedding models are limited to single words as inputs; thus, we extracted keywords $K$ from the reformulated query using the appropriate strategy from Section 3.4 based on the dataset. Upon obtaining our keywords, we assign weights $w$ to them based on their frequency across the reformulated query, facilitating effective measurement of each keyword's importance (since keywords will be repeated across the reformulated query). This results in a refined query where each keyword is paired with its weight, forming a structured representation as a list of tuples expressed as:

$$x_R^w = ((K_1, w_1), (K_2, w_2), \ldots, (K_n, w_n))$$

In the equation, $x_R^w$ represents the reformulated word representation that will be used as input into the static word embedding model, and (K, w) is the keyword and its corresponding weight.

## 3.6 Zero-shot text classification

We explore zero-shot text classification using both contextual embedding and static word embedding models (Table 3) to leverage the distinct advantages of each embedding type. Contextual embeddings capture the overall meaning of the reformulated query, while static word embeddings allow for finer-grained analysis of the individual keywords in the reformulated query. Both approaches are explained as follows:

- **Contextual Embedding Models:** To perform zero-shot classification, we use the contextual embedding models to obtain embeddings for the reformulated query $x_R^s$ and each potential class label. Next, we compute the cosine similarity (a measure of closeness between embeddings) between $x_R^s$ and each class. The class with the highest cosine similarity is assigned to $x_R^s$ (See Figure 1).

- **Static Word Embedding Models:** First, we compute the embedding of each class label. For class labels with single words, we obtain their representation directly from the embedding model, while we average the representations of the constituent words within the label if it is a phrase. We also obtain the representation for each keyword $K$ in the $x_R^w$. To determine the best matching topic, we compute the cosine similarity between each keyword $K$ in $x_R^w$ and each class label $y$, subsequently multiplying this similarity by the corresponding weight $w$ of $K$. These weighted similarities across all keywords in $x_R^w$ are then aggregated to yield cumulative scores for each class label. Finally, the class label with the highest accumulated score is assigned to $x_R^w$. See Algorithm 1 for details.

**Table 2: Dataset Classification Summary**

| Dataset | Classification Type | # Classes | # Test | Class Labels |
|---|---|---|---|---|
| AG News | News Topic | 4 | 7.6K | politics & government, sports, business & finance, technology |
| DBPedia | Wikipedia Topic | 14 | 70K | companies, schools & university, artists , athletes, politics, transportation buildings & structures, mountains & rivers & lakes, villages, animals, plants & trees, albums, films, books & novels & literature |
| Yahoo! Answers | Web QA Topic | 10 | 60K | society & culture, science & mathematics, health, education & reference,internet & computers, sports, business & finance, entertainment, family & relationships, politics & government |
| Yummly | Cuisine Type | 20 | 7.9K | Cajun creole, Jamaican, Chinese, French, Vietnamese, Filipino, Irish, Thai, Indian, Southern United States, Moroccan, Greek, Italian, Japanese, Mexican, Korean, Russian, Spanish, British, Brazilian |
| TagMyNews | News Topic | 7 | 6.5K | sports, business, entertainment, United States, politics & government, health, science & technology |
| Ohsumed | Disease Topic | 23 | 4K | bacterial infections, virus diseases, parasitic diseases, neoplasms, musculoskeletal diseases, digestive system diseases, stomatognathic diseases, respiratory tract diseases, otorhinolaryngology diseases, nervous system diseases, eye diseases, urologic male genital diseases, female genital diseases, pregnancy complications, nutritional & metabolic diseases, cardiovascular diseases, hemic & lymphatic diseases, neonatal diseases, skin & connective tissue diseases, endocrine diseases, immunologic diseases, environmental disorders, animal diseases, pathological conditions |

---

**Algorithm 1** : Classification using Static Word Embedding Models

---

**Require:** $x_R^w$ (List of tuples: keyword and weight)
**Require:** word_embed (Model: returns an embedding vector)
**Require:** $Y$ (List of class labels)
**Ensure:** Class label with the highest cosine similarity
  Initialize **class_scores** as an empty dictionary
  **for** each $y \in Y$ **do**
    $y\_score \leftarrow 0$
    **for** each $(K, w) \in x_R^w$ **do**
      $K\_embed \leftarrow$ word_embed$(K)$
      $y\_embed \leftarrow$ word_embed$(y)$
      $sim \leftarrow$ cosine_similarity$(K\_embed, y\_embed)$
      $W\_sim \leftarrow sim \times w$
      $y\_score \leftarrow y\_score + W\_sim$
    **end for**
    $class\_scores[y] \leftarrow y\_score$
  **end for**
  $best\_class \leftarrow$ key with maximum value in $class\_scores$
  **return** $best\_class$

---

## 4 EXPERIMENTAL SETUP

### 4.1 Datasets

For a comprehensive understanding of our study and to ensure a thorough examination of textual nuances across diverse domains, we evaluated QZero on six distinct publicly available text classification datasets. These include the AG News articles [36], and the DBPedia factual knowledge base [17], which contains a summary of Wikipedia extracts. We also utilize the Yahoo! Answers community-driven knowledge exchange [36], and the Yummly

**Table 3: Embedding Models Evaluated**

| Model | Embedding Type |
|---|---|
| Word2Vec [21] | static word |
| GloVe [22] | static word |
| FastText [2] | static word |
| All-mpnet-base-v2 [29] | contextual |
| text-embedding-3-small (GPT-3-small) | contextual |
| text-embedding-3-large (GPT-3-large) | contextual |

dataset [13] of recipes from various regional cuisines, obtained from Kaggle's What's cooking challenge [4]. Additionally, we employed the TagMyNews dataset [13][5], containing news from RSS feeds as adopted by [18], and the Ohsumed corpus [10][6], a collection of medical abstracts about various diseases. We used labels similar to [15, 7, 13, 18], which we show in Table 2. We report the classification accuracy averaged over three runs on the test sets. We summarize all dataset statistics in Table 2.

### 4.2 Zero Shot Models and Baselines

We evaluated the impact of the QZero pipeline on embedding models by comparing their performance utilizing the original input versus reformulated queries. Six different static word and contextual embedding models (See Table 3) were tested.

- Zero-shot classification via Contextual Embeddings: The approach is similar to the contextual embedding classification

---

[4]https://www.kaggle.com/competitions/whats-cooking/data
[5]https://github.com/AIRobotZhang/STCKA
[6]https://disi.unitn.it/moschitti/corpora.htm

method described in Section 3.6. The only difference is that instead of the reformulated query, we compare the cosine similarity between the embeddings of the original text input and class labels to achieve zero-shot classification.

- Zero-shot classification via Static Word Embeddings: The key difference between the baseline static word embedding approach in Section 3.6 and this baseline approach is the absence of weights in the input to be classified. This means we compared each class label to the average vector representation of words in the original input text.

We accessed the text-embedding-3 small and large models through the OpenAI API[7] in March 2024. while the All-mpnet-base-v2 model was accessed via hugging face[8]. We describe the All-mpnet base-v2 model, text-embedding-3 small and large models as All-mpnet, GPT-3-small, and large, respectively, throughout the manuscript. Unlike the All-mpnet base-v2 model, with a 512-token limit, OpenAI models can handle longer inputs. To ensure consistency and address the 512-token limit across all models, we employed the GPT-2 tokenizer[9] when reformulating queries.

## 4.3 Retrieval Models

To retrieve supporting categories, we explored two methods: a dense and a sparse retriever. The sparse retriever utilized the BM25 [26] algorithm implemented in Pyterrier [19] with default settings. For the dense retriever, we employed the Contriever [11] model, trained specifically on Wikipedia passages. The BEIR [30] framework enabled indexing and retrieval for the Contriever model.

## 5 RESULTS AND DISCUSSION

### 5.1 Effect of Retrieval Augmentation on Zero-shot Performance

Table 4 showcases the benefits of the retrieval augmentation pipeline (QZero), demonstrating performance improvements across all model sizes, especially in News topic classification datasets. In the TagMyNews dataset, the smallest model (Word2vec) experienced a significant 13.00% boost in accuracy, while even the largest model (GPT-3-large) saw a 6.61% increase. Similarly, in the AG News dataset, all models achieved a minimum accuracy gain of 4.17%, except for GPT-3-large, which exhibited a slight 1.57% drop. This drop in GPT-3-large's performance suggests potential noise introduced by uninformative categories in the reformulated query.

QZero enhances smaller models to achieve performance comparable to larger models without QZero. In TagMyNews, a QZero-enhanced Word2vec outperformed GPT-3-large and GPT-3-small (with original input) by substantial margins of 3.56% and 9.27%, respectively. Similarly, in the AG News dataset, Word2vec outperformed GPT-3-small by 3.4% while achieving similar accuracy with GPT-3-large. This is particularly valuable for scenarios with limited computational resources or tight financial constraints, where utilizing OpenAI's expensive embeddings might not be feasible.

Furthermore, QZero enriches the original input with useful context information that may be outside of the model's training data.

For example, in the Ohsumed disease topic dataset, GPT-3-large and Word2vec (a model with limited medical knowledge) achieved a minimum of 5.00% increase in accuracy. On the Yummly recipe datasets, the static word embedding models achieved a boost as high as approximately 38.00%. In addition, even the All-mpnet-base-v2 model, also lacking training data in the culinary domain, improved by 17.54% on the Yummly recipes. This is impressive considering the limited medical and culinary domain information present in the general Wikipedia corpus, which was the only Knowledge corpus for QZero. These results highlight the promising potential of QZero as a cost-effective solution for domain adaptation challenges.

Our results also demonstrate that retrieval augmented query reformulation is effective for classifying topics outside of the model's training data. By transforming the input query into a knowledge space the model is more familiar with, reformulation bridges the gap between the model's knowledge and unfamiliar topics. This enhances the model's adaptability, allowing it to handle a wider range of tasks and domains without extensive retraining. Additionally, this reformulation provides support for a more effective representation, leading to models that are better equipped to tackle unseen data or generalize across domains.

Interestingly, we observed that across datasets, QZero rarely hurts the performance of the static word embedding models, and in cases where it does, the performance decline is typically minimal, under 1.00%. However, the impact on larger models is more diverse, especially for common evaluation sets like DBpedia and Yahoo Answers. Here, QZero's effect can vary in magnitude, with the largest decrease observed in GPT-3-large. This disparity might be linked to the training data used for these larger models. The Yahoo Answers dataset, for instance, was part of the training data for All-mpnet-base-v2[10]. Similarly, GPT-3 models are trained on massive-scale datasets that might already contain the information encoded in the reformulated queries. As a result, applying QZero for such datasets might introduce redundancy or contradict existing knowledge within these larger models, leading to performance drops.

### 5.2 Effect of Dense vs. Sparse Retrieval Models

The findings presented in Table 4 highlight QZero's robustness, showcasing its compatibility with dense and sparse retrievers. The BM25 (sparse) retriever performs better on the Ohsumed dataset, which contains lengthy documents that most dense neural retrievers might struggle with. Interestingly, the BM25 retriever also outperforms the Contriever (dense) retriever on the Yummly dataset. This could be because Contriever's training data was not well-suited for the specific domain of recipes in Yummly. In contrast, the Contriever (dense) retriever excels in tasks related to News topics, general QA, and Wikipedia topics, domains likely aligning better with its training data. These findings demonstrate how QZero adapts to the complementary strengths of each retrieval method, ultimately broadening its applicability across various use cases.

---

[7]https://platform.openai.com/docs/guides/embeddings
[8]https://huggingface.co/sentence-transformers/all-mpnet-base-v2
[9]https://huggingface.co/docs/transformers/en/model_doc/gpt2

[10]training datasets can be found: via:https://huggingface.co/sentence-transformers/all-mpnet-base-v2

**Table 4: Model Performance Summary across Six Datasets. QZero CTV (Contriever) represents the intervention of QZero using the dense retriever, and QZero BM25 represents the intervention of QZero using the sparse retriever.**

| Models | TagMyNews (%) | | | AG News (%) | | | Ohsumed (%) | | | Yummly (%) | | | DBpedia (%) | | | Yahoo (%) | | |
|---|---|---|---|---|---|---|---|---|---|---|---|---|---|---|---|---|---|---|
| | Base | +QZero BM25 | +QZero CTV | Base | +QZero BM25 | +QZero CTV | Base | +QZero BM25 | +QZero CTV | Base | +QZero BM25 | +QZero CTV | Base | +QZero BM25 | +QZero CTV | Base | +QZero BM25 | +QZero CTV |
| Word2Vec | 46.37 | +11.75 | +13.00 | 65.24 | +9.84 | +11.06 | 18.35 | +5.00 | +3.22 | 4.36 | +38.23 | +21.41 | 69.00 | -0.72 | +1.37 | 42.63 | +5.09 | +6.17 |
| GloVe | 38.84 | +17.52 | +18.06 | 60.59 | +9.46 | +10.78 | 12.64 | +9.37 | +5.61 | 16.62 | +16.30 | +4.08 | 61.72 | +12.46 | +14.96 | 46.72 | -0.98 | +0.30 |
| FastText | 44.30 | +14.73 | +15.04 | 69.96 | +8.33 | +8.93 | 8.66 | +12.23 | +11.23 | 8.26 | +26.89 | +17.69 | 70.86 | -0.59 | +1.33 | 37.36 | +9.19 | +11.18 |
| All-mpnet | 49.85 | +8.86 | +9.21 | 76.28 | +4.65 | +4.61 | 46.27 | +0.38 | -2.42 | 27.10 | +17.54 | +6.84 | 78.02 | +2.03 | +4.05 | 52.65 | -6.55 | -6.55 |
| GPT-3-small | 50.10 | +10.60 | +12.23 | 72.90 | +4.57 | +4.17 | 34.55 | +0.15 | -2.69 | 37.47 | -2.75 | +0.28 | 74.16 | -3.02 | -0.04 | 50.49 | -5.94 | -7.63 |
| GPT-3-large | 55.81 | +4.85 | +6.61 | 76.93 | -1.07 | -1.57 | 37.45 | +5.53 | +2.59 | 54.36 | -9.26 | -13.32 | 78.50 | -5.13 | -3.30 | 53.28 | -3.54 | -3.44 |

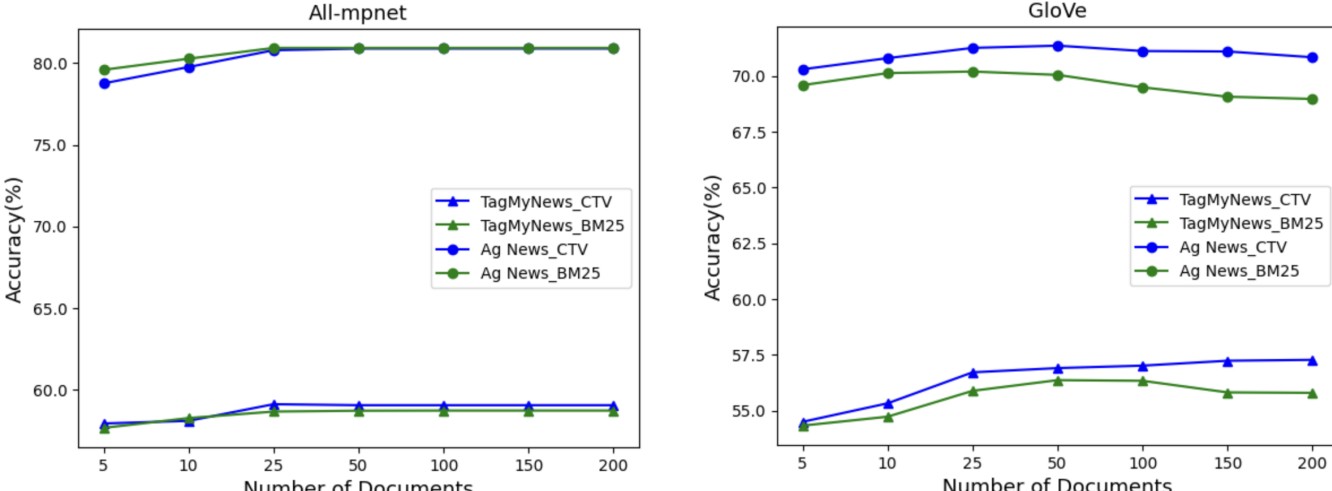

**Figure 2: Performance of All-mpnet and GloVe embedding models on selected datasets as a function of the number of document categories retrieved. (Contriever) CTV represents the dense retriever, and BM25 represents the sparse retriever.**

## 5.3 Analysis of QZero's Outputs

Table 5 shows that QZero's capabilities go beyond simply classifying queries. By leveraging Wikipedia's knowledge base, it generates insightful details for each query, including relevant categories and keywords. These insights illuminate the context of the query and verify its connection to specific topics, enriching our understanding of the model's predictions. For instance, for the query "Today's executives address Lauer and Vieiera's exit buzz," QZero identifies categories like "American television shows," confirming and validating the entertainment ground truth. Furthermore, QZero's outputs can help in understanding the query's focus. Even without prior sports knowledge, QZero's outputs reveal the sports-related nature of the query "Martinez leaves bitter like Roger Clemens did almost exactly eight years earlier, Pedro Martinez has left the Red Sox apparently bitter about the way he was treated by management," aiding in interpreting the model's predictions.

Our analysis of incorrect predictions reveals a few key sources of error. In some cases, the predicted topic and the annotated ground truth might differ, but both could be valid interpretations of the query's meaning. Alternatively, the ground truth itself could be inaccurate. For example, in Table 5, the query "Why do I not walk correctly when I have sinusitis? Your equilibrium could be 'off' due to your sinusitis, which could cause problems with your inner ear" clearly has no connection to the ground truth, "Business & Finance." Other errors stem from retrieving irrelevant categories or limitations of the embedding model itself. By understanding these nuances, we can leverage the model's capabilities to interpret its predictions, refine our evaluation methods, and ultimately enhance model accuracy.

## 5.4 Effect of Number of Retrieved Documents

Figure 2 shows how QZero's performance changes with respect to the number of documents retrieved for reformulating queries. This applies to both GloVe and All-mpnet embedding models. We see a trend across all datasets: as more documents are retrieved initially, QZero's performance improves. However, once the number of retrieved documents exceeds 50, the accuracy plateaus in the case of the All-mpnet model due to the fixed number of maximum tokens

**Table 5: QZero gives useful insights about a query.**

| Input Query | Returned Categories | Top Keywords | Ground Truth | Predicted Topic |
|---|---|---|---|---|
| Martinez leaves bitter like Roger Clemens did almost exactly eight years earlier, Pedro Martinez has left the Red Sox apparently bitter about the way he was treated by management | 1999 Major League Baseball season, New York Yankees postseason, Boston Red Sox postseason, American League Championship Series, 1999 in sports in Massachusetts,... | (players, 414), (baseball, 120), (people, 76), (sports, 67), (births, 39), (postseason, 38), (coaches, 35), (competitions, 33), (managers, 30), (season, 28) | Sports | Sports |
| Today's executives address Lauer and Vieiera's exit buzz | 2010s American television talk shows, 2014 American television series debuts, 2016 American television series endings, English-language television shows, First-run syndicated television programs in the United States,... | (television, 122), (people, 64), (films, 62), (series, 53), (shows, 40), (episodes, 36), (language, 30), (news, 27), (books, 22) (births, 20) | Entertainment | Entertainment |
| soy sauce, salt, pork tenderloin, hoisin sauce, toasted sesame seeds, sugar, dry sherry | Beijing cuisine, Pork dishes, Chinese condiments, Chinese sauces, Vietnamese cuisine, Beijing cuisine,... | (Chinese, 70), (Cantonese, 13), (Philippine, 13), (Japanese, 13), (Pork, 12), (American, 12), (Oregon, 11), (Indonesian, 10), (Korean, 9), (Thai, 7) | Chinese Cuisine | Chinese Cuisine |
| Ky. Company Wins Grant to Study Peptides (AP) AP - A company founded by a chemistry researcher at the University of Louisville won a grant to develop a method of producing better peptides, which are short chains of amino acids, the building blocks of proteins | Companies based in Suffolk County, Biotechnology companies of the United States, Biotechnology companies established in 2005, 2005 establishments in New York (state), Biotechnology companies of the United States, Research support companies ... | (companies, 132), (biotechnology, 26), (century, 18), (establishments, 16), (people, 14), (pharmaceutical, 14), (alumni, 13), (care', 10), (faculty, 9), (women, 8) | Science & Technology | Business |
| Why do I not walk correctly when I have sinusitis? Your equilibrium could be 'off' due to your sinusitis, which could cause problems with your inner ear. | 'Cancer, Head and neck cancer, Otorhinolaryngology, Inflammations, Diseases of inner ear, Hygiene, Rhinology, Diseases of the ear and mastoid process ... | (diseases, 17), (ear, 17), (disorders, 14), (medicine, 11), (head, 9), (neck, 9), (mastoid, 8), (syndromes, 4), (cancer, 3), (virus, 2) | Business & Finance | Health |
| Some People Not Eligible to Get in on Google IPO Google has billed its IPO as a way for everyday people to get in on the process, denying Wall Street the usual stranglehold it's had on IPOs. Public bidding, a minimum of just five shares, an open process with 28 underwriters - all this pointed to a new level of public participation. But this isn't the case. | Initial public offering, Corporate finance, Types of auction, Stock exchanges in India, Stock market terminology ... | (companies, 27), (law, 20), (establishments, 9), (market, 7), (offering, 5), (scandals, 5), (finance, 4), (securities, 4), (litigation, 4), (stock, 3), | Science & Technology | Business |

the model can take as input. On the other hand, the accuracy of the GloVe embedding model starts to decline. This suggests that there's a point of diminishing return where retrieving more documents starts to include irrelevant ones. Retrieving too many documents dilutes the pool of relevant categories that appropriately describe the input query, ultimately reducing QZero's effectiveness.

## 6 CONCLUSIONS AND FUTURE WORK

Embedding models present an effective solution for zero-shot text classification. However, the conventional method of retraining or fine-tuning the entire model to improve embedding quality proves costly, especially in rapidly evolving domains. QZero addresses this challenge by offering a training-free solution that improves the embedding quality. It achieves this through the adoption of a retrieval augmented query reformulation, which is considerably more cost-effective to update. Our experiments and results demonstrate that QZero significantly boosts classification accuracy across a wide range of embedding models, ranging from smaller options like Word2Vec to larger models such as OpenAI's text embeddings.

Furthermore, QZero enhances the performance of smaller models to match their larger counterparts.

Beyond improving classification accuracy, QZero incorporates relevant information from Wikipedia articles, providing valuable insights into the context of queries and validating their pertinence to specific topics. This dual functionality strengthens the rationale behind model predictions, leading to more reliable and trustworthy classification outcomes. In summary, QZero improves the performance of embedding-based zero-shot classifiers while maintaining their simplicity, paving a promising path for more efficient and interpretable zero-shot classification techniques.

While QZero shows promise, there are limitations that require future exploration. First, the current query reformulation process for contextual embedding models could benefit from further refinement. Secondly, there might be cases where the model's knowledge is simply insufficient, and query reformulation alone would not be enough to improve performance. Lastly, our experiments focused on embedding models. Investigating whether query reformulation or other forms of retrieval augmented learning can benefit other models, such as those for natural language inference or generation, would be an interesting avenue to explore.

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
