# OpenReview forum: "Retrieval Augmented Zero-Shot Text Classification"
_ACM.org/SIGIR/ICTIR/2024/Conference — ICTIR 2024_

### Official Review · Reviewer_94Kz · 2024-05-10

**Rating:** -1
**Confidence:** 4

**Objective Part Of Review:**

(1) Is the problem clearly stated?

This paper aims to enhance zero-shot text classification by augmenting the contextual information of query text through retrieval.

(2) Are the methods clearly described?

This paper presents a retrieval-based query reformulation method for zero-shot text classification. It utilizes a retriever model (BM25 or Dense Retriever) to recall relevant document categories for the query text, concatenates these categories as the reformulated query, and employs existing word embedding models (e.g., word2vec) to obtain embeddings for the reformulated query and potential classes. The prediction is based on the class with the highest similarity score between them.


(3) Are the results clearly stated?

Yes, the paper provides corresponding results on six datasets and conducts some hyperparameter studies.

(4) Are the various claims in the paper supported?

Most claims are supported.

(5) Is each concept and notation properly defined before it is used?

Yes.

(6) Can the abstract and the introduction be understood before one has read and understood the rest of the paper?

Yes.

(7) Did you spot any contradictions or other signs that something is wrong?

No.

(8) Is work that is directly relevant or even competing with the work in this paper cited?

Yes.

(9)Provide falsifiable evidence for your criticism:

My main concern is that the results indicate the proposed method tends to decrease performance for stronger  contextual embedding models, like GPT-3-large, across most datasets. Does this suggest the proposed method is only suitable for weaker word embedding models, such as word2vec?

Another concern is that while ERATE targets text similarity tasks, it can also be adapted for zero-shot text classification. I believe the proposed method should compare against these stronger baselines.

**Subjective Part Of Review:**

(1) Did you find the paper easy to read and understand?

Yes, the paper is easy to follow.

(2) Do you find the problem relevant?

Yes.

(3) Do you find the methods original?

While this paper may offer original contributions to the field of zero-shot classification, the proposed framework lacks originality in broader domains, such as retrieval-augmented generation.

(4) Do you find the results interesting?

I gained little insight from the results. Most of them align with expectations.

(5) Do you think that others in the ICTIR community will be interested in this work?

Have some interests.

---

### Official Review · Reviewer_CtNi · 2024-05-17

**Rating:** -1
**Confidence:** 3

**Objective Part Of Review:**

This article introduces a novel training-free augmentation method called QZero, which is used to enhance the performance of zero-shot text classification. QZero reframes queries by retrieving supporting categories from Wikipedia, thereby enhancing the quality of embeddings used for zero-shot classification.

* The model used in this paper's method is weak, and the drop in GPT3 effects is not enough to suggest that the method supports a large model.

**Subjective Part Of Review:**

* The paper employs some baselines, such as Word2Vec and other static word embedding models, which are outdated. It also fails to incorporate some of the latest open-source large models, such as LLaMa, for experimentation.

* From Table 4, it is evident that large models like GPT-3 do not perform as well in many classification tasks after using RAaG (Retrieval-Augmented Generation) as they do in their base form. This suggests that the method proposed by the authors has limitations, particularly when applied to large models.

* The paper primarily focuses on text classification based on text embedding methods. However, for large models, it is more common to use generative approaches for classification, which involves directly inputting the candidate categories and the text to be classified into the large model and allowing the model itself to generate the category. Additionally, RAaG is more suitable for generative methods.

---

### Official Review · Reviewer_yiGT · 2024-05-17

**Rating:** 1
**Confidence:** 4

**Objective Part Of Review:**

This paper studies the task of zero-shot text classification. The authors argue that simple embedding-based methods embed the query independently and cannot encode contextual information, hindering performance. The authors propose to use a retriever to retrieve contextual information to refine the original query. The motivation is clear, and the experimental results are good.

Question:

In Table 3, the authors mark text-embedding-3-small as a "GPT-3-small" model, and text-embedding-3-large as a "GPT-3-large" model. which may not be correct - there is no evidence indicating the underlying model. OpenAI never mentions the underlying models of the two embedding APIs.

**Subjective Part Of Review:**

The presentation is generally good and I think the ICTIR community will be interested.

---

### Official Review · Reviewer_8nip · 2024-05-19

**Rating:** 1
**Confidence:** 3

**Objective Part Of Review:**

The problem was formulated in a sufficiently clear way, perhaps I missed some important aspects, since it seems to me that the problem is the one addressed through one of the many Query Expansion techniques such as Relevance Feedback. If so then the problem seems to be the one given more by the use of Word Embedding techniques than by the ambiguity of the queries, the latter being widely addressed, known and perhaps "solved".

The methodology is described quite well. The authors have made an effort to make it clear even to non-experts, particularly those who, like me, know IR very well but know WE less well and deal less frequently with classification. If the work were accepted and there was margin I would like the authors to give a few more definitions on the basic concepts used.

**Subjective Part Of Review:**

The work is quite understandable thanks to a linear writing and free of great confusion. The ease of reading has certainly favored my appreciation of the relevance of the problem also due to the self-interest cultivated for a long time. This is also why I did not find the proposed methods particularly original since it seemed to me to recognize many other similar methods described in the literature dedicated to Query Expansion. For limitations of personal knowledge I cannot say with certainty that the methods are original in the context of the WE and automated classification by neural networks and the like. However, I think others in the ICTIR community can find food for discussion and reflection.

---

### Meta-Review · Area_Chair_7B6A · 2024-06-03

**Recommendation:** Reject
**Confidence:** 4

**Metareview:**

This paper propose a retrieval augmentation to enhance zero-shot classification.  The setup is to use frozen text-embeddings from models like Word2Vec / OpenAI text-embeddings API for classification. Under this setup, the authors argue that when the input text is short, the embedding models often lack knowledge, and cannot embed the text close to its class labels. The authors therefore propose to retrieve similar wikipedia passages (BM25/Contriever), add the wikipedia passages' categories to the input text, and re-embed the text for classifcation. Experiments show that this method significantly improve models like Word2Vec and FastText, but the gains on newer/strong embedding models such as OpenAI text-embeddings-3 is limited.

The paper is clearly written. The motivation is very intuitive. The method is simple yet can boost the quality of weak embedding models.

However, the current version of the paper is not sufficiently convincing due to the following reasons:
1) On many datasets, the proposed method cannot outperform vanilla OpenAI text-embeddings-3-large. The complexity added from this method is not justified.
2) Related to 1, the proposed seems to only benefit weak models like Word2Vec, and it needs to be explained why people would chose to use those older models over newer embedding models.
3)  Using frozen text-embeddings is one approach for zero-shot classification. E.g., one could prompt LLMs in a generative manner to generate the class labels. The authors should add these common zero-shot / few-shot classifcation baselines.

Therefore, we do not think the paper is ready for ICTIR. We encourage the authors to revise the paper, in particular adding stronger baselines to justify the claims.